# Prediction of Sea Level Nonlinear Trends around Shandong Peninsula from Satellite Altimetry

**DOI:** 10.3390/s19214770

**Published:** 2019-11-02

**Authors:** Jian Zhao, Ruiyang Cai, Yanguo Fan

**Affiliations:** 1College of Ocean and Space Information, China University of Petroleum (East China), Qingdao 266580, Chinaygfan@upc.edu.cn (Y.F.); 2Laboratory for Marine Mineral Resources, Qingdao National Laboratory for Marine Science and Technology, Qingdao 266071, China

**Keywords:** sea level anomaly, prediction, Shandong Peninsula coast, nonlinear trends, satellite altimetry

## Abstract

Sea level change is a key indicator of climate change, and the prediction of sea level rise is one of most important scientific issues. In this paper, the gridded sea level anomaly (SLA) data from satellite altimetry are used to analyze the sea level variations around Shandong Peninsula from 1993 to 2016. Based on the Complete Ensemble Empirical Mode Decomposition (CEEMD) method and Radial Basis Function (RBF) network, the paper proposes an improved sea level multi-scale prediction approach, namely, CEEMD-RBF combined model. Firstly, the multi-scale frequency oscillatory modes (intrinsic mode functions (IMFs)) representing different oceanic processes are extracted by CEEMD from the highest frequency to the lowest frequency oscillating mode. Secondly, RBF network is used to establish prediction models for various IMF components to predict their future trends, and each IMF is used as an input factor of the RBF network separately. Finally, the prediction results of each IMF component with RBF network are reconstructed to obtain the final predictions of sea level anomalies. The results shows that CEEMD is particularly suitable for analyzing nonlinear and non-stationary time series and RBF network is applicable for regional sea level prediction at different scales.

## 1. Introduction

Under the changing climate, mean sea level rise is mainly responsible for more frequent coastal flooding around the world in recent decades and can drastically impact coastal communities [1]. Therefore, modelling sea level change is an important technique and is used widely to evaluate and study shorelines and climate changes in coastal areas [2,3,4]. Unlike the global sea level change, regional sea level patterns may be more complicated as their influential factors are mixed with those due to dynamical ocean responses to natural climate variability, so their physical processes are not clear. During the 20th century, the global mean sea level has risen at a rate of about 1.7 mm/year [5], and it will continue to rise through the 21st century due to enhanced greenhouse gas emissions [6,7]. Tide gauges measure sea level change relative to a benchmark located on coastlines with a coarse spatial resolution. Therefore, these observations always include coastal effects such as land movements or wave and wind surges, while satellite altimeters measure sea level change with respect to the center of mass of the Earth, providing nearly global sampling of sea level with high measuring accuracy [8,9,10]. Historical tide gauge observations indicate that the global mean sea level rise rate is 1.7 ± 0.2 mm/year over 1900–2009 [1]. The global mean sea level rise rate from satellite altimetry is 3.2 ± 0.4 mm/year over 1993–2009 [11], and 3.1 ± 0.4 mm/year over 1993–2017 (http://sealevel.colorado.edu/content/2018rel1-global-mean-sea-level-time-series-seasonal-signals-removed).

Sea level change is not geographically uniform and shows great regional differences, while regional mean seal level rise trends can be significantly different from the global one. Therefore, accurate estimates of regional sea level change are important for controlling and predicting its potential impact on coastal regions [12,13,14,15]. China’s coastal areas are one of the most vulnerable areas under climate change scenarios. With regards to the historical record, the reported local coastal mean sea level rise with tide gauge data has been estimated to be in the range of 1.4–2.9 mm/year since 1950 [16]. In the satellite altimeter era, the regional sea level in China coast exhibits a higher rise rate than the global mean value [10]. With satellite altimetry data, the average sea level change in the East China Sea is about 3.9 mm/year from 1992 to 2009 [17], and in South China Sea it reaches up to 5.5 mm/year from 1993 to 2009 [9], while the average rate of sea level rise in China Sea is 4.64 mm/year from 1993 to 2012 [10]. Currently, there are more studies about the seasonal signals and trends of sea level variations in South China Sea and East China Sea and less studies about the Bohai Sea and Yellow Sea [9,10,14,15]. The sea level change study in the China medium–high latitude offshore areas needs to be further enhanced in order to fully identify the characteristics in China offshore seas.

Recently, increasing efforts have been made towards understanding past sea level trends and producing more accurate projections of future sea level rise on the regional scale, which are the basis for implementing appropriate adaptation measures for coastal areas [14,15,16,17,18,19,20]. A series of time–frequency analysis techniques, such as the Fourier analysis, wavelet analysis, and other spectral analysis methods, are applied to the study of sea level change modelling, in which the sea level time series are decomposed into deterministic trend terms, cycle terms, and residual random noise series. The trend terms and cycle terms can be extrapolated and the residual random noise can be fitted and predicted linearly/nonlinearly [21,22,23,24,25,26,27].

The artificial neural network is a well-known method already used in a variety of oceanography and climate applications [25,26,27]. The Radial basis function (RBF) neural network is a feedforward neural network that consists of three layers, namely, the input, hidden, and output layers [12,28]. An RBF network features relatively high computing speed and good nonlinear reflecting capability and can approximate a nonlinear function at arbitrary precision [26,28]. Using the radial basis function (RBF) neural network, Zhao et al. (2019) forecasted the residual terms of sea level anomalies (SLA) time series and proved the reliability of RBF network in the short-term prediction of sea level variations [26].

The empirical mode decomposition (EMD) method has been successfully applied in a variety of disciplines (climatology, hydrology, meteorology, biosciences, economics, etc.) [29,30,31,32,33,34,35,36]. The EMD method can process non-stationary and nonlinear signals without the need of predetermining the specific form of the trend terms [30]. The Ensemble Empirical Mode Decomposition (EEMD) method is also used to obtain an independent estimate of sea level trends and ensure that the estimate is not sensitive to the analysis technique employed [34,35]. Recently, several studies utilized the EEMD or EMD methods to analyze sea level change using tide gauge data in different regions [24,31,32,33,36]. If there are abnormal points in the original signal, the uneven distribution of extreme points is easy to lead to mode mixing in the EMD decomposition [27,33]. In order to suppress mode mixing of the EMD method, Wu et al. [34] proposed the Complete Ensemble EMD (CEEMD) method. The CEEMD method applies noise-assisted analysis to effectively suppress mode mixing through adding white noise to the original signal to smooth out anomalous points, and the uniform distribution of the white noise spectrum is used to automatically distribute the signals at different time scales to the appropriate reference scale [34,35]. There are no studies about CEEMD method on regional sea level variations in coastal zone using altimeter observations.

In this paper, the CEEMD method is first applied to estimate the sea level trends around Shandong Peninsula over a 24-year period from 1993 to 2016 using the gridded sea level anomaly (SLA) data combined from several altimetry missions. Further, the RBF neural network is applied to predict the sea level nonlinear trends of intrinsic mode functions (IMFs) decomposed by CEEMD around Shandong Peninsula. Firstly, the multi-scale frequency oscillatory modes (intrinsic mode functions, IMFs) representing different oceanic processes around Shandong Peninsula are extracted by CEEMD from the highest frequency to the lowest frequency oscillating mode. The remaining non-oscillating mode (residual) is the sea level trend. Secondly, the RBF network is used to establish prediction models for various IMFs and the residual to predict their future trends, and each IMF and residual is used as an input factor of the RBF network separately. Thirdly, the prediction results of each IMF and the residual with RBF network are reconstructed to obtain the final sea level prediction results. Finally, the prediction results of pure RBF network without the CEEMD decomposition is used to compare with the results of the CEEMD-RBF combined model. At last the CEEMD-RBF combined model is used to predict the sea level change around Shandong Peninsula in the next 10 years.

## 2. Data and Methodology

### 2.1. Data

In this study, the merged sea level anomaly (SLA) time series derived from all available altimeter missions (TOPEX/Poseidon, ERS-1/2, Jason-1/2, Envisat, GFO and Cryosat, etc.) provided by AVISO (Archiving, Validation and Interpretation of Satellite Oceanographic Data) are used. The SLA is calculated as the difference of the sea surface height relative to a mean sea surface model and the mean sea surface height model from 1993 to 1999 is taken as the reference. The SLA data is weekly gridded at a resolution of 0.25 × 0.25 degrees and available from January 1993 to June 2016. All standard geophysical, environmental and orbital corrections have been applied, including ionospheric delay and dry/wet tropospheric effects, solid Earth tides, ocean tides, and pole tides, the loading effects of the ocean tides, sea state bias, the inverse barometer response of the ocean, and instrument corrections, etc. Detailed information about the corrections can be found at the AVISO website (https://www.aviso.altimetry.fr/en/home.html). The study area is the coastal area around Shandong Peninsula (34°00′ N–38°00′ N, 119°00′ E–123°00′ E), as shown in Figure 1.

The weekly SLA time series around Shandong Peninsula after regional averaging over period of January 1993 to June 2016 is shown in Figure 2. From Figure 2, it can be seen that the sea level variability from 1993 to 2016 generally shows a significant rising trend, with the linear trend being about 2.49 ± 0.32 mm/year. The sea level has obvious seasonal and inter-annual variation characteristics; it reaches high value in summer and falls to low value in winter and spring, and the minimum appears in 1994 and maximum appears in 2001.

### 2.2. Complete Ensemble Empirical Mode Decomposition (CEEMD) Method

The main steps of CEEMD decomposition are as follows [34,35]:

Step 1: A pair of Gaussian white noise n(t) with opposite amplitudes and the mean of zero are randomly added to the original signal h(t), and two new signals are obtained:(1)$$\{\begin{array}{c} y_{1}(t)=h(t)+n(t)\\ y_{2}(t)=h(t)-n(t) \end{array}.$$
where $y_{1}(t)$, $y_{2}(t)$ are the time series signals after adding positive and negative white noise respectively; n(t) is the white noise signal; and h(t) is the original signal.

Step 2: the new signals of $y_{1}(t)$ and $y_{2}(t)$ are decomposed by EMD separately, and each signal gets a set of IMF components. The average value of each set of IMFs are recorded as IMF1 and IMF2, respectively.

Step 3: the average of the sum of IMF1 and IMF2 are calculated respectively, then the decomposition results of CEEMD are:(2)$$s_{1}=(\mathrm{IMF}1+\mathrm{IMF}2)/2.$$

Step 4: repeat the above steps n times to obtain the IMF component $s_{i}(t)$ and the residual component f(t):
(3)$$s_{i}(t)=\frac{1}{2n}\sum_{j=1}^{n}\left(s_{ji}(t)+s_{-ji}(t)\right),$$
(4)$$f(t)=\frac{1}{2n}\sum_{j=1}^{n}\left(f_{j}(t)+f_{-j}(t)\right).$$

Then the target signal h(t) can be expressed as the sum of (m−1) IMF components and the residual component, namely:(5)$$h(t)=\sum_{i=1}^{n}s_{i}(t)+f(t),$$
where $s_{i}(t)$ is the *i*_th_ IMF component after decomposition, 1≤i<m; f(t) is the residual component after decomposition; $s_{ji}(t)$ is the *i*_th_ IMF of the *j*_th_ signal, 1≤j≤n; n is the number of groups to add the white noise.

## 3. Results and Discussion

### 3.1. Sea Level Nonlinear Trends Analysis

CEEMD analysis performed to investigate the nonlinear sea-level variability using the SLA time series from 1993 to 2016 around Shandong Peninsula (Figure 2), produced eight IMFs in the data-set considered, corresponding to seven period cycles and a long-term trend item, as shown in Figure 3. All the IMFs which account for different temporal scales of variability were identified. The period cycles mainly include 2 months, 3 months, 12 months, 28 months, 37 months, 66 months, and 168 months, which basically correspond to the conventional theoretical astronomical cycles. Among them, the first three cycles correspond to intra-seasonal variability and annual cycles: the 28-month cycle is a quasi-two-year atmospheric oscillation; the 37-month cycle and 66-month cycle are consistent with the 3–7 year El Niño Southern Oscillation (ENSO) cycle, and the 168-month cycle may be a tidal cycle similar to the 14-year cycle in astronomy.

It was found that the most significant cycle of sea level variations around Shandong Peninsula is the annual cycle, followed by the 2–3 months cycle, and the 2.3–3.1 years cycle are also relatively remarkable, but the semi-annual cycle is not obvious. The primary cycle is the annual cycle and the 2–3 months cycle is also remarkable, indicating that the sea level variations around Shandong Peninsula are remarkably characterized by seasonal changes. This is mainly because Shandong Peninsula is located within the East Asian monsoon area of the North Temperate Zone, where it is usually hot and rainy in summer, cold and rainless in winter. The typical monsoon climate is the main cause of the sea level seasonal variations [37]. The 2.3–3.1 years cycle (the inter-annual sea level variations) reflects generally the climatic interaction between ocean and atmosphere, the most important of which are caused by ENSO, through its impact on the global water cycle [38].

After CEEMD analysis a long-term trend term is left, as shown in Figure 3 bottom panel (*h*), which contains the component whose time scale is larger than the length of the data, reflecting the long-term “natural trend” of the signal. The linear regression on the residuals during time period 1993–2016 gives a trend of 2.54 ± 0.24 mm/year. This value is in good agreement with the sea-level rise around Shandong Peninsula estimated by Guo et al. for the time from 1993 to 2012, namely 2.37 mm/year [10]. The long-term trends obtained by CEEMD analysis feature nonlinearity and reflects the nonmonotonic of the sea level change in a detailed manner, which cannot be obtained by the previous methods used for linear trend analysis.

### 3.2. Prediction of Sea Level Nonlinear Trends

#### 3.2.1. Construction of CEEMD-RBF Combined Model

The prediction of sea level change based on the CEEMD-RBF combined model mainly includes the CEEMD decomposition of SLA time series and prediction of IMF components based on RBF neural network. Firstly, CEEMD is used to extract the signals with different characteristics and scales of SLA time series and reduce the influence of noise, then various IMF components of varying time scale or trend items in the original signals are obtained, the period and trends implied in sea level change are determined eventually, as shown in Section 3.1. Secondly, the RBF network is used to establish prediction models for different scale IMF components to obtain their future trends. Finally, the prediction results of all IMF components are reconstructed to obtain the prediction results of the original SLA time series.

Considering the sea level change around Shandong Peninsula has obvious seasonal variation characteristics, we use the corresponding weekly SLA time series to construct the data modeling and establish the RBF network through moving the average of original data. Sea level anomalies from altimeters covering 1993–2016, a total of 864 weekly SLA data are divided into two parts. The first 698 weekly data covering the period of 1993–2011 are used for the training procedure, whereas the remaining 167 weekly data covering 2012–2016 are utilized for the testing procedure.

The selection of appropriate input variables is critical for a successful neural network modeling. Zhao et al. have demonstrated the computation of statistical analysis to determine the appropriate input variables of RBF network [26]. Following routine procedures for the selection of the best neural network suited to the sea level data, the appropriate model structure for the neural network applied in RBF architectures are determined [26]. Several trainings are performed to determine the number of hidden layers and the number of neurons in the hidden layers that provide the best testing performance. Furthermore, different numbers of hidden layer neurons and spread constants are also examined for the RBF network models.

Herein, the optimal RBF network prediction model is designed as follows: the input layer neurons are 12 and the output layer is 1 (i.e., if the input layer is the SLA of first week of January from 1993 to 2004, then the output layer is the SLA of first week of January in 2005), and the number of hidden layer neurons is 20. The maximum training number of RBF network is 50,000 times, the maximum error of the system is less than 0.001. The RBF network modeling process is implemented in Matlab (R 2017b).

In the prediction of CEEMD-RBF combined model, the IMF l–8 components obtained by CEEMD decomposition are predicted by RBF neural network, respectively. The results of the predictions of IMF l–8 components for the training period (take the SLA data from 2007 to 2011 as an example) compared with the observed sea level values are illustrated in Figure 4. For the training period, the prediction results of the high-frequency components of IMF l (Figure 4a) are relatively poor, but the overall amplitude and direction of the prediction results are consistent with the sample data. The main reason for the poor prediction of IMF l is that the high-frequency components of sea level change are complex, and there are a large number of mutations and extreme points. Compared with the long-term trends predicting, neural network model is relatively inferior to the extreme values prediction. From IMF 2 to IMF 8, the prediction effect of RBF network gradually becomes better as each IMF component tends to stable and the prediction accuracy also increases. Generally, the estimations obtained from the RBF network agree well with the observed sea level data after CEEMD decomposition, indicating the RBF network works excellently for sea level trends predicting.

The results of the predictions of IMF l–8 components for the testing period (2012–2016) compared with the observed sea level values are illustrated in Figure 5. For the testing period, the prediction results of the high-frequency components of IMF l (Figure 5a) are also not ideal, but the overall amplitude and direction of the predictions are consistent with the sample data. From IMF 2 to IMF 8, the prediction accuracy of RBF network significantly increases as each IMF component tends to stable, indicating the RBF network has good performance for sea level trends predicting. When studying the long-term trends of sea level change, the high-frequency sub-annual oscillation signal is usually removed from the original time series firstly [37]. Moreover, the seasonal cycles of sea level change around Shandong Peninsula are mainly composed of annual cycle and the not ideal fitting effect of IMF l component has little impact on the long-term trends prediction of sea level variations.

#### 3.2.2. Analysis of Prediction Results

To demonstrate the capability of the CEEMD-RBF combined model to predict sea level anomalies around Shandong Peninsula, a comparison is also made between the CEEMD-RBF combined model and the pure RBF network model, which is a traditional model directly used to predict the sea level anomalies without CEEMD decomposition. Similar to the description in Section 3.2.1, The RBF network model consists of two parts, namely, the training procedure and the testing procedure. Eventually, the RBF (12, 22, 1) model provides the optimal result for sea level prediction around Shandong Peninsula. In order to avoid deviations from the original time series caused by noise reduction processing of CEEMD, all prediction results of IMF l–8 components using RBF network models are superimposed and reconstructed to obtain the final prediction results of CEEMD-RBF combined model. The prediction accuracy is computed using the correlation coefficient (R), the mean absolute error (MAE) and the root mean square error (RMSE) as follow [25,26]:(6)$$R=\left[\frac{\sum_{i=1}^{n}\left(h(t_{i})-\bar{h}(t_{i}))(\hat{h}(t_{i})-\bar{\hat{h}}(t_{i}\right))}{\sqrt{\sum_{i=1}^{n}{\left(h(t_{i})-\bar{h}(t_{i})\right)}^{2}\sum_{i=1}^{n}{\left(\hat{h}(t_{i})-\bar{\hat{h}}(t_{i})\right)}^{2}}}\right],$$
(7)$$MAE=\frac{1}{n}\sum_{i=1}^{n}\left|\hat{h}(t_{i})-h(t_{i})\right|,$$
(8)$$RMSE=\sqrt{\frac{1}{n}\sum_{i=1}^{n}{\left[\hat{h}(t_{i})-h(t_{i})\right]}^{2}},$$
where $h(t_{i})$ is the input SLA time series at the time $t_{i}$, $\hat{h}(t_{i})$ is the prediction of the $h(t_{i})$, n is the number of the $\hat{h}(t_{i})$, and $\bar{h}(t_{i})$ and $\bar{\hat{h}}(t_{i})$ are the average of the original data and prediction of SLA, respectively.

The statistics of the optimal RBF network model and the CEEMD-RBF combined model for validation are presented in Table 1. The optimal RBF network model gives the minimum RMSE of 28.30 mm, minimum MAE of 22.04 mm and the maximum R of 0.92. The RBF network model cannot surpass the CEEMD-RBF combined model in the sea level prediction around Shandong Peninsula. The RMSE obtained from the CEEMD-RBF combined model is reduced by about 18%, and MAE reduces more than 16% compared with the values obtained from the RBF network model. Cheng et al. used the tide gauge data and satellite altimetry data to predict the sea level change around England, and the minimum RMSE of the predicted results and observed data was 43.9 mm [39]. Compared with the results of Cheng et al., the overall prediction effect of our study is ideal. The multi-scale predictions of RBF network based on the different characteristics of the original signal decomposed by CEEMD is better than the prediction of pure RBF network. This difference can be attributed to the fact that the SLA time series are first smoothed and de-noised to guarantee prediction accuracy when using the CEEMD-RBF combined model to predict the sea level change, especially if the investigated phenomenon is nonlinear.

Figure 6 presents the comparison of predicted results using pure RBF network and CEEMD-RBF combined model for testing data from 2012–2016 around Shandong Peninsula. From Figure 6 it is found that the overall trends of the prediction results from both RBF network and CEEMD-RBF combined methods are in good agreement with the original SLA time series. However, the predictions of the RBF network are more prone to extreme deviation while the predictions of the CEEMD-RBF combined model are more consistent with the original data. At the same time, data that deviates significantly from the original SLA time series during the prediction process of RBF network, that is, the points where the prediction error is relative larger are more likely to appear at the extreme points of sea level anomalies. It should be noted that although the future trends of sea level change have been simulated and predicted by learning the sea level historical variations from modeling process of the RBF network. There are a large number of mutations and extreme points due to the complexity of sea level change, the predictions of extreme values in the SLA data are relatively poor compared with the predictions of the long-term trends.

#### 3.2.3. Prediction of CEEMD-RBF Combined Model for the Next 10 Years

Considering the validity of the SLA data, we used the CEEMD-RBF combined model and the IMF 2–7 components and the trend term of IMF 8 (residual) are used to predict the sea level change around Shandong Peninsula for the next 10 years, as shown in Figure 7. In Figure 7, the blue solid line is the SLA time series from 1993 to 2016 around Shandong Peninsula and the linear fitting results show that the rate of sea level rise to be about 2.49 mm/year. The red solid line is the prediction results of the CEEMD- RBF combined model and the linear fitting result of sea level rise reach up to 3.0 mm/year by 2025. According to the 2018 China Sea Level Bulletin, the rate of sea level rise of China coast is about 3.3 mm/year in the past 30 years, and it is predicted that the sea level in the Yellow Sea will rise by 80–160 mm in the next 30 years [40]. In our study, the satellite altimetry data is used to study the sea level change around Shandong Peninsula and the prediction results are consistent with the above conclusions.

#### 3.2.4. Application of CEEMD-RBF Combined Model in the Yangtze River Estuary Coastal Area

The Yangtze River Estuary (YRE) coastal area is a region located between the Yellow Sea and the East China Sea. Rising sea levels would severely threaten the costal zones of the YRE due to high density population and fast growing economies. In order to test the prediction effect of the combined model in different geographic region, we also use the CEEMD-RBF combined model to predict the sea level change in the YRE coastal area in the next 10 years (Figure 8). In Figure 8, the blue solid line is the SLA time series from 1993 to 2015 in the YRE coastal area and the linear rate of sea level rise is about 3.28 mm/year. The red dotted line is the prediction of the CEEMD-RBF combined model and the linear rate of sea level rise will reach up to 3.42 mm/year by 2025. The prediction results in the YRE coastal area are consistent with the conclusions of the 2018 China Sea Level Bulletin [40]. It is shown that the sea level anomaly time series in coastal areas can be predicted successfully using the CEEMD-RBF combined model.

## 4. Conclusions

The sea level anomaly time series from altimeter around Shandong Peninsula over a period spanning two decades (from 1993 to 2016) are used to establish a CEEMD-RBF combined model for long-term trend predictions research. The CEEMD method is used to analyze the cycles, trends, and nonlinear variations of sea level change and an RBF neural network was applied to predict SLA data capturing long-term trends, periodic oscillations, and stochastic components. CEEMD analysis produced eight IMFs in the data-set considered, corresponding to seven period cycles and a long-term trend item. The period cycles mainly include 2 months, 3 months, 12 months, 28 months, 37 months, 66 months, and 168 months, which basically correspond to the conventional theoretical astronomical cycles. The linear regression of the residuals during the time period 1993–2016 gives a trend of 2.54 ± 0.24 mm/year. The residual shows a nonlinear trend with a declined slope around Shandong Peninsula in recent years. The CEEMD-RBF predictions of sea level anomalies are significantly more accurate than the corresponding pure RBF predictions around Shandong Peninsula. Finally, the CEEMD-RBF combined model is used to predict the sea level variations around Shandong Peninsula from 2016 to 2025.

When using the CEEMD-RBF combined model, the SLA time series are firstly decomposed by CEEMD and periodic items and trend items with different time–frequency characteristics are obtained. Then the RBF network is used to establish prediction models for each IMF component and eventually the prediction results of all IMFs are reconstructed to obtain the final prediction results. It can be seen from the prediction results of each IMF component that the more stable the time series, the better the prediction effect of the RBF network. The prediction of CEEMD-RBF combined model is based on the smoothing and de-noising of the SLA time series and is more reasonable, thus the prediction accuracy is significantly improved.

Because of its advantages, such as without any priori assumptions about the nature of the generating processes, neural network models are suitable for the non-stationary and nonlinear time series prediction. However, the ability of neural network to capture extreme values of sea level change time series is relatively limited. In future studies, we need to pay more attention to the prediction of extreme values and the return period of sea level anomalies, and the latest neural network models such as deep learning models can also be introduced to the study of sea level change modelling.

## Figures and Tables

**Figure 1 sensors-19-04770-f001:**
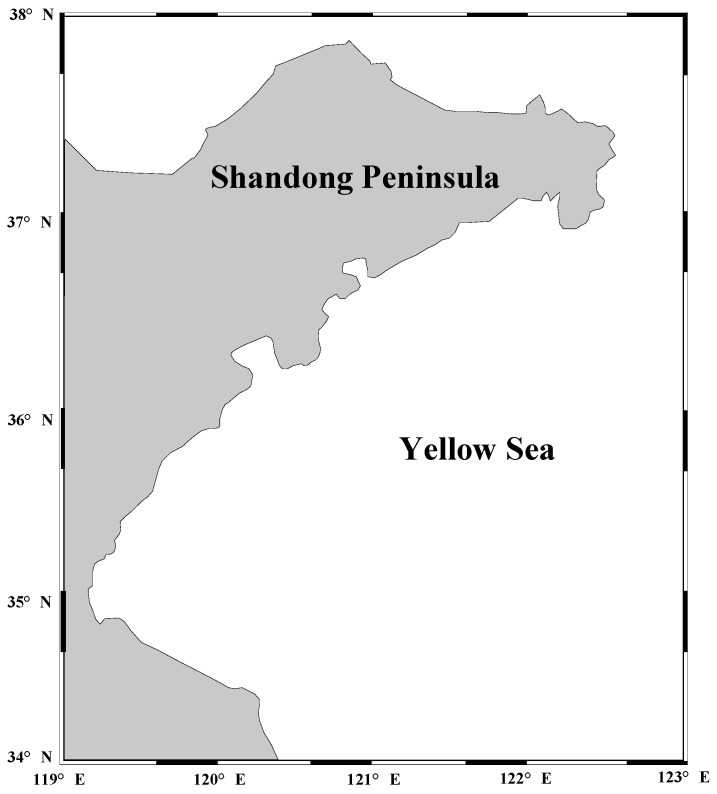
Study area around Shandong Peninsula.

**Figure 2 sensors-19-04770-f002:**
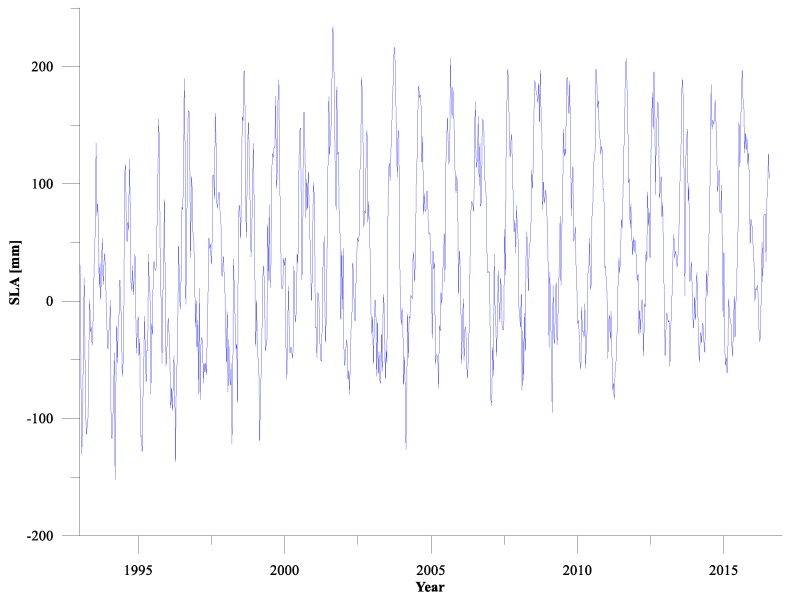
Weekly sea level anomaly (SLA) time series from 1993 to 2016 around Shandong Peninsula.

**Figure 3 sensors-19-04770-f003:**
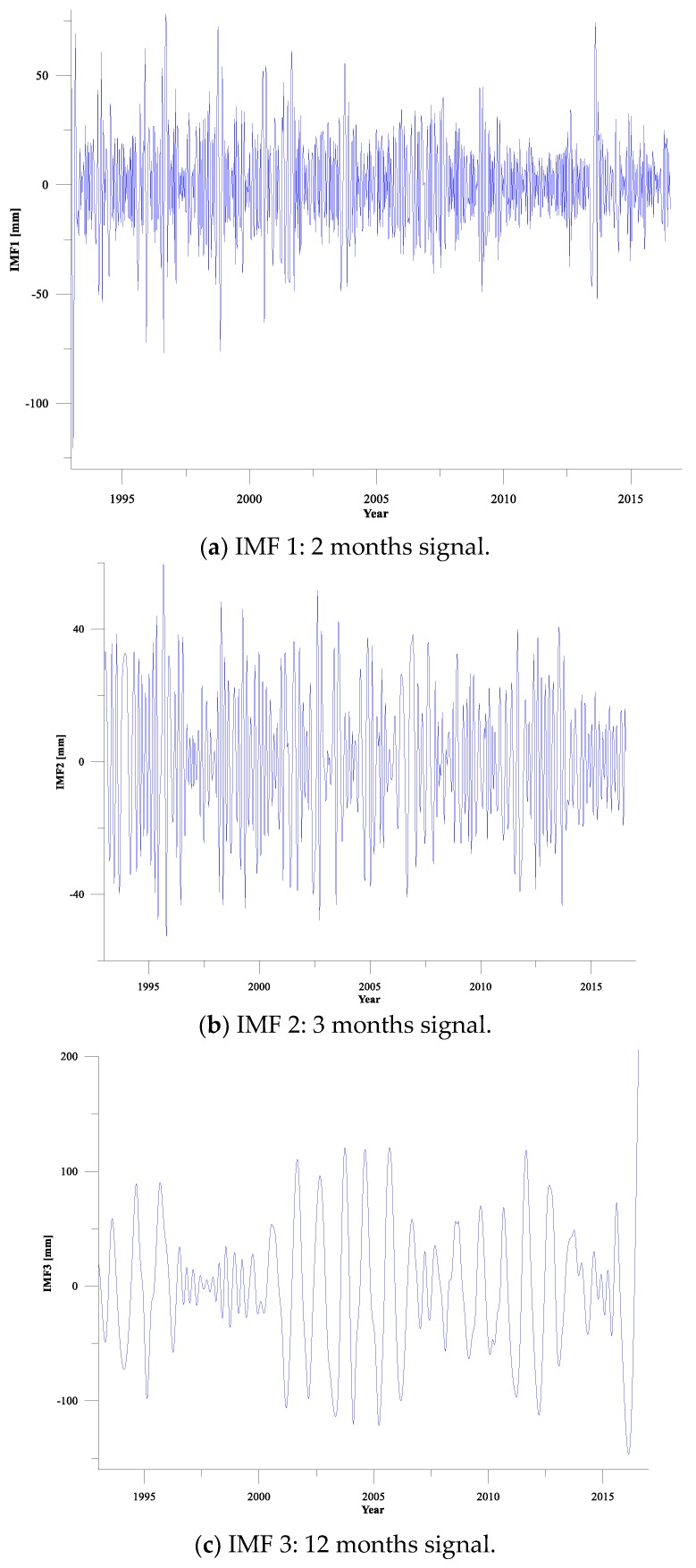
Complete Ensemble Empirical Mode Decomposition (CEEMD) decomposition results of SLA time series around Shandong Peninsula (Panels (**a**–**g**) represent the IMFs with cycles respectively, Panel (**h**) represents the residual.).

**Figure 4 sensors-19-04770-f004:**
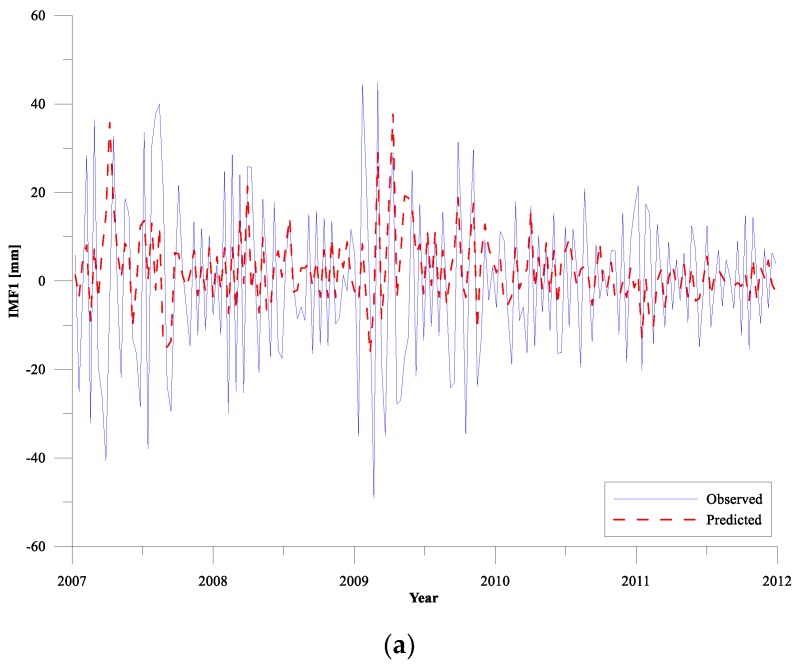
Comparison of predictions of IMF components by CEEMD decomposition for training period with optimal RBF network (Panels (**a**–**g**) represent the IMFs with cycles respectively, Panel (**h**) represents the residual.).

**Figure 5 sensors-19-04770-f005:**
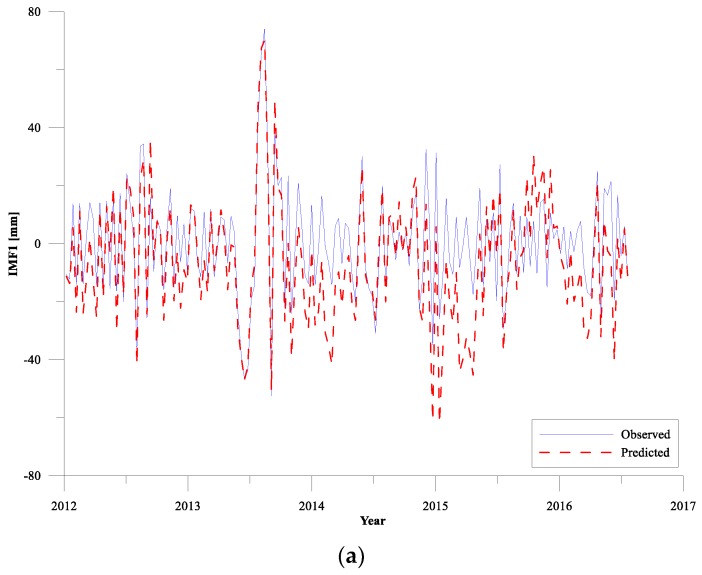
Comparison of predictions of IMF components by CEEMD decomposition for testing period with optimal RBF network (Panels (**a**–**g**) represent the IMFs with cycles respectively, Panel (**h**) represents the residual.).

**Figure 6 sensors-19-04770-f006:**
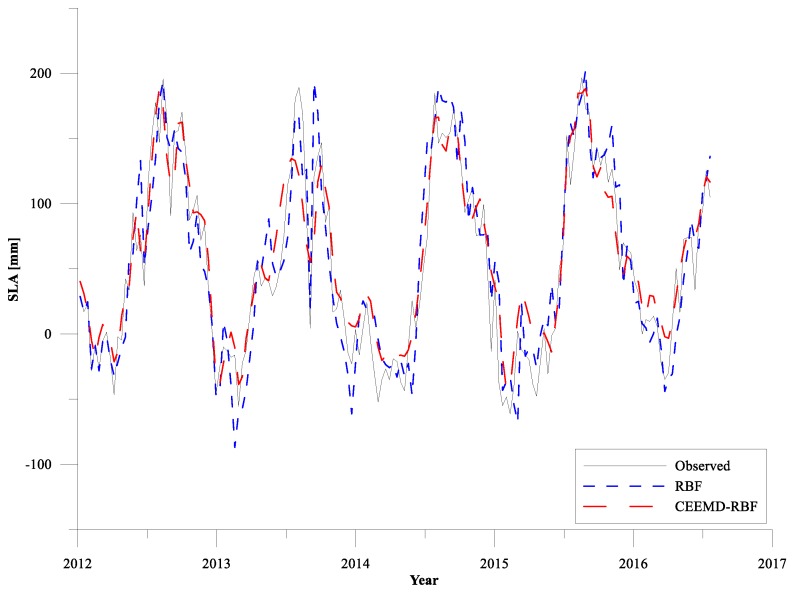
Observed and estimated sea level anomalies derived by the optimal RBF model and CEEMD-RBF combined model during the testing period.

**Figure 7 sensors-19-04770-f007:**
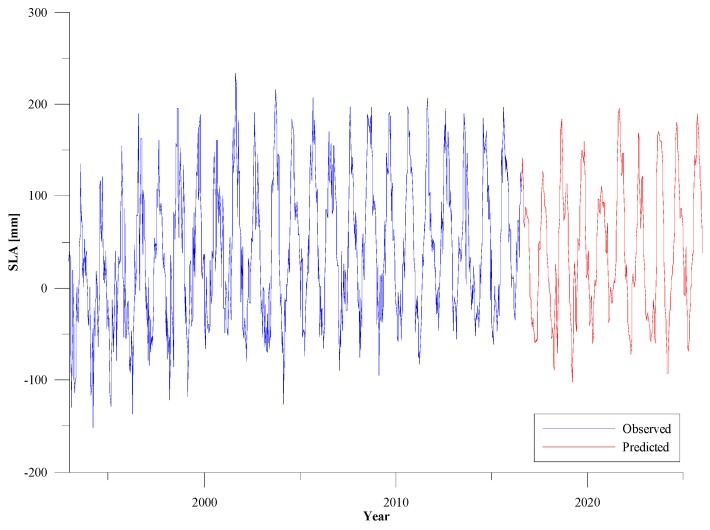
Estimated SLAs with CEEMD-RBF combined model from 2016 to 2025 around Shandong Peninsula.

**Figure 8 sensors-19-04770-f008:**
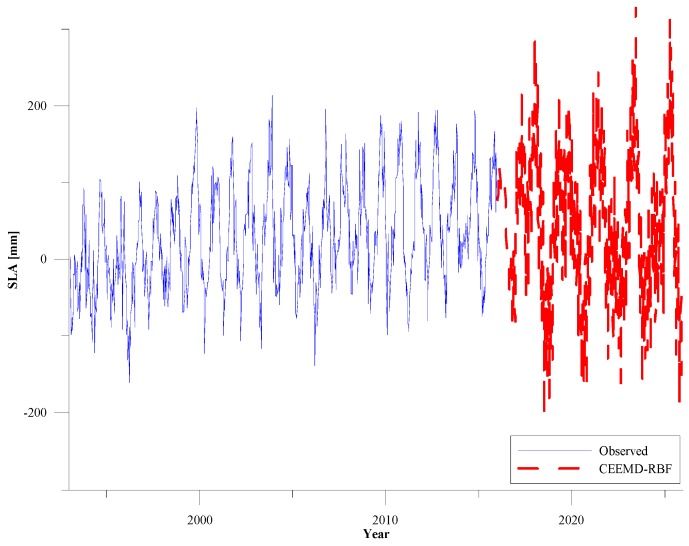
Estimated SLAs with CEEMD-RBF combined model from 2016 to 2025 in the Yangtze River Estuary coastal area.

**Table 1 sensors-19-04770-t001:** Statistics of prediction results computed for RBF and CEEMD-RBF combined model.

Model	R	MAE (mm)	RMSE (mm)
RBF	0.92	22.04	28.30
CEEMD-RBF	0.95	18.35	23.12

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
