# Peer review of "Prediction of Sea Level Nonlinear Trends around Shandong Peninsula from Satellite Altimetry"

_sensors, 2019, doi:10.3390/s19214770_

Round 1
Reviewer 1 Report
The paper proposes an improved sea level rise prediction approach and combined model. It is a well written and organised paper representing a good contribution for Sensor journal.
My only main comment refers to the results presented and conclusions: although the manuscript would like to deal with sea level variations around Shandong Peninsula (China), I think that results and outcomes achieved by the authors would have a stronger significance and impact if they can be confirmed in different geographic regions. My suggestion for authors is thus to adding another (at least) study case, selecting a different geographic region and replicate there their analyses.
Author Response
Response 1: According to the comments of the reviewer, the Yangtze River Estuary (YRE) coastal area is chosen as a different geographic region and the CEEMD-RBF combined model is also used to predict the sea level change in the YRE coastal area in the next 10 years. The predictions of the CEEMD-RBF combined model is satisfactory (page 23, line 362-373; page 24, line 374-375 of the revised version).
Reviewer 2 Report
The paper presents an analysis of SLA variations, trend identification and prediction algorithms
for the marine area around Shandong Peninsula in China. The main asset of the paper is that it
demonstrates that the combined use of CEEMD analysis and RBFs thorough a properly designed
neural network approach, manages to provide overall good agreement for the testing years and
through that it can be used for projection into the future for SLA analysis.
In general the paper is worthy for publication, subject to some minor changes identified below.
1) line 12: omit "the"
2) line 29: omit "the" before mean sea level
3) line 31: rephrase as "..., modeling sea level change
4) line 33: rephrase as "as their influential factors are mixed with those"
5) line 34: rephrase as "variability, so their physical processes are not clear"
6) line 46: replace "and regional" with "while regional"
7) line 47: Either use "Therefore, the accurate estimation" or "Therefore, accurate estimates"
8) Line 49: Either "The Chinese coastal areas" or "China coastal areas:
9) Line 62: omit "As"
10) Line 73: Correct as Zhao et al. (2019)
11) Line 83: Correct as "....if there are abnormal...."
12) Line 85: Correct as "Wu et al. [34] proposed"
13) Line 86: Do you mean "through"??
14) Line 112: Rephrase as " as the difference of the sea surface height relative to a mean sea surface model.
In the present study the mean sea surface model from 1993 to 1999 is taken as the reference". Add which MSS
has been used as reference. Is it the CNES MSS?
15) Line 114: The authors use weekly gridded SLAs from AVISO with a spatial resolution of 0.25x0.25 degrees.
This means that given the area under study, which has an effective sea area of 3.5x3.5 degrees, that the
available number of observations is ~196-200. Is this resolution sufficient to study the SLA variations?
Moreover, given the complexity of the coastline, is the resolution sufficient to give useful
information about coastal areas?
16) Line 124: ....is shown
17) Line 125: correct as "with the linear trend being about..."
18) Figure 12. The label of the y-axis is not proper. Put the units in brackets as [mm]
19) Lines 131-132: What do the authors mean by "a constant standard deviation of amplitude". Why is the
standard deviation of the added noise mixed with the amplitude of the signal (SLAs)? Do they mean that the
standard deviation of the noise added is some percentage of the signal amplitude?
20) Section 2.2 is not well written. The EEMD principles are not outlined in step 2 so the reader
cannot grasp how the IMFs Cij are determined. There is no information on how the IMFs are estimated, how
the residuals are determined and how the noise of the observations is iteratively added. More details and
the explicit mathematical expressions are needed.
21) In Section 3.1 the authors present the results of the CEEMD decomposition and identify the IMFs. First,
it would be nice to relate the IMFs in the order they appear in Figure 3 with the identified periodic cycles.
That means to add to the captions of each sub-figure the period that the specific IMF refers to.
Also, the seasonal cycles depicted in the first two panels of Figure 3 do not convey much information. Maybe
a zoom to a specific year would add more information.
According to Eq. (3), if we add all the IMFs and the residual part we should reconstruct the original signal.
The SLA variations are within +-200mm, but the axis of the IMFs and their variations is within +-100mm, hence
quite large. Are the IMFs values in [mm] or some other unit?
Can the authors provide some information on why the annual cycle show reduced amplitude between ~1996 to 2000?
Finally, how is ECEMD compared to EOF and PCA analysis? What are the benefits that it offers? In my view,
the analysis and results presented in Section 3.1 can be achieved very well with EOF/PCA.
In general, the presentation of this section is not very convincing.
22) Line 223. What do you mean by "default value". If the authors used the neural toolbox by Matlab, then
they should explicitly say so and indicate which variables they have not changed and to which model of the
toolbox.
23) Figure 5. The IMF1 component for the testing period seems better predicted compared to the IMF1
component for the training period. How can this be explained?
Author Response
The paper presents an analysis of SLA variations, trend identification and prediction algorithms for the marine area around Shandong Peninsula in China. The main asset of the paper is that it demonstrates that the combined use of CEEMD analysis and RBFs thorough a properly designed neural network approach, manages to provide overall good agreement for the testing years and through that it can be used for projection into the future for SLA analysis.
In general the paper is worthy for publication, subject to some minor changes identified below.
1) Line 12: omit "the"
2) Line 29: omit "the" before mean sea level
3) Line 31: rephrase as "..., modeling sea level change
4) Line 33: rephrase as "as their influential factors are mixed with those"
5) Line 34: rephrase as "variability, so their physical processes are not clear"
6) Line 46: replace "and regional" with "while regional"
7) Line 47: Either use "Therefore, the accurate estimation" or "Therefore, accurate estimates"
8) Line 49: Either "The Chinese coastal areas" or "China coastal areas:
9) Line 62: omit "As"
10) Line 73: Correct as Zhao et al. (2019)
11) Line 83: Correct as "....if there are abnormal...."
12) Line 85: Correct as "Wu et al. [34] proposed"
13) Line 86: Do you mean "through"??
Response 1: Thank you very much for the typos comments in the paper and all of the typos from 1) to 13) are all corrected.
14) Line 112: Rephrase as "as the difference of the sea surface height relative to a mean sea surface model. In the present study the mean sea surface model from 1993 to 1999 is taken as the reference". Add which MSS has been used as reference. Is it the CNES MSS?
Response 2: A 7-year (1993-1999) mean SSH is subtracted from the time series to eliminate the unknown geoid. Indeed, the period of reference was 7years in older version (from 1993 to 1999) and it is now 20 years (from 1993 to 2012). This takes into account the variations of the oceans in the last years.
The SLAs are computed from the difference of the instantaneous SSH minus a temporal reference. This temporal reference can be a Mean Profile in the case of repeat track or a gridded MSS when the repeat track cannot be used. The MSS used in the SLA products is MSS_CNES_CLS11, referenced [1993, 2012].
15) Line 114: The authors use weekly gridded SLAs from AVISO with a spatial resolution of 0.25x0.25 degrees. This means that given the area under study, which has an effective sea area of 3.5x3.5 degrees, that the available number of observations is ~196-200. Is this resolution sufficient to study the SLA variations? Moreover, given the complexity of the coastline, is the resolution sufficient to give useful information about coastal areas?
Response 3: As a consequence of the time and space sampling of the satellite altimetry described above, gridded altimetry data are limited to solve structures larger than 37 km. This is a clear limitation of this dataset to study the coastal region since spatial and temporal scales are often shorter in shallow waters. But compared to the coarsely scattered tide gauge stations in the study area around Shandong Peninsula, the quality and quantity of altimetry data is still better.
16) Line 124: ....is shown
17) Line 125: correct as "with the linear trend being about..."
Response 4: The typos from 16) to 17) are corrected.
18) Figure 2. The label of the y-axis is not proper. Put the units in brackets as [mm]
Response 5: The labels of the y-axis from Figure 2 to Figure 8 are all corrected as [mm].
19) Lines 131-132: What do the authors mean by "a constant standard deviation of amplitude". Why is the standard deviation of the added noise mixed with the amplitude of the signal (SLAs)? Do they mean that the standard deviation of the noise added is some percentage of the signal amplitude?
20) Section 2.2 is not well written. The CEEMD principles are not outlined in step 2 so the reader cannot grasp how the IMFs Cij are determined. There is no information on how the IMFs are estimated, how the residuals are determined and how the noise of the observations is iteratively added. More details and the explicit mathematical expressions are needed.
Response 6: According to the comments of the reviewer, The CEEMD principles (2.2) are reorganized and rewritten (page 4, line 131-136; page 5, line 137-152 of the revised version).
21) In Section 3.1 the authors present the results of the CEEMD decomposition and identify the IMFs. First, it would be nice to relate the IMFs in the order they appear in Figure 3 with the identified periodic cycles. That means to add to the captions of each sub-figure the period that the specific IMF refers to. Also, the seasonal cycles depicted in the first two panels of Figure 3 do not convey much information. Maybe a zoom to a specific year would add more information.
According to Eq. (3), if we add all the IMFs and the residual part we should reconstruct the original signal. The SLA variations are within +-200mm, but the axis of the IMFs and their variations is within +-100mm, hence quite large. Are the IMFs values in [mm] or some other unit?
Can the authors provide some information on why the annual cycle show reduced amplitude between 1996 to 2000?
Finally, how is CEEMD compared to EOF and PCA analysis? What are the benefits that it offers? In my view, the analysis and results presented in Section 3.1 can be achieved very well with EOF/PCA. In general, the presentation of this section is not very convincing.
Response 7: According to the comments of the reviewer, the periods that the specific IMF refers to are added to the captions of each sub-figure of Figure 3. In order to convey much information of IMF1~2, zooming to a specific year (1998-1999) are shown in the word file.
We confirm that all the IMFs and the residual are reconstructed to obtain the original signal. Because the IMFs signal is randomly distributed, positive and negative will offset. The unit of the IMF values in [mm] is correct.
The reason of the annual cycle shows reduced amplitude between 1996 and 2000 is that the El Niño Southern Oscillation (ENSO) event occurred during this period, resulting in the amplitudes of 28 months and 66 months are very large.
CEEMD is an adaptive method for analyzing nonlinear and non-stationary signals and has the ability to reconstruct the original signal perfectly. Compared with EOF and PCA analysis, the signals extracted by CEEMD method feature a high signal to noise ratio and represent different physical meanings.
22) Line 223. What do you mean by "default value". If the authors used the neural toolbox by Matlab, then they should explicitly say so and indicate which variables they have not changed and to which model of the toolbox.
Response 8: The training and predicting of RBF network model is programmed and implemented in Matlab, and we don’t use the neural toolbox of Matlab to process the RBF network modeling. In order not to cause ambiguity, the "default value" is deleted in the revised version.
23) Figure 5. The IMF1 component for the testing period seems better predicted compared to the IMF1 component for the training period. How can this be explained?
Response 9: The first 698 weekly data covering the period of 1993–2011 are used for the training procedure of the RBF network, whereas the remaining 167 weekly data covering 2012–2016 are utilized for the testing procedure. For the training period, the prediction results of the high-frequency components of IMF l are relatively poor, and we only show a part of the data of IMF 1 (184 weekly data) from 2007 to 2011 in Figure 4 a.
The reason of the IMF1 component for the testing period seems better predicted compared to the IMF1 component for the training period is mainly that the testing period is shorter than the training period, the number of the mutations and extreme points of the testing period is less than the training period.
Response 10:
According to the comments of the reviewer, some other typos are also corrected and a new reference [40] is updated (page 27, line 517-518 of the revised version).
